# RES-BENCH: REASONING SKILL-AWARE REASONING DIAGNOSTIC EVALUATION BENCHMARK FOR MATH REASONING

## ABSTRACT

Large Language Models (LLMs) have recently demonstrated strong performance on mathematical reasoning tasks, often evaluated solely by their ability to produce the correct final answer. However, this evaluation paradigm fails to capture whether models genuinely follow sound reasoning processes or rely on spurious shortcuts. In this paper, we introduce **Res-Bench**, a first fine-grained evaluation dataset for measuring the mathematical reasoning abilities of LLMs not only on final correctness but also on **step-level reasoning quality** and **reasoning skill alignment**. Specifically, Res-Bench consists of 3271 test samples, primarily focused on math problems aligned with Chinese middle- and high-school curricula, provided in English. Each test case is annotated by GPT-4 and verified by human experts with its decomposition into intermediate reasoning steps, and mapped to explicit reasoning skills. Based on Res-Bench, we further conduct extensive evaluation with a multi-dimensional evaluation protocol that measures: (1) final answer accuracy, (2) consistency and validity of intermediate steps, and (3) mastery over the required reasoning skills. Our experimental results across several state-of-the-art LLMs reveal that while models can often achieve high answer-level accuracy, **their step-level reasoning exhibits significant inconsistencies and frequent misalignment with targeted reasoning skills.** Our findings highlight the necessity of moving beyond final-answer evaluations and toward process-based assessment, providing deeper insights into LLMs' reasoning capabilities. [1]

## 1 INTRODUCTION

Recently, mathematical datasets have become critical benchmarks for evaluating the reasoning capabilities of LLMs, indicating that LLMs like GPT-4 and Claude-3 (Achiam et al., 2023; The) can solve increasingly complex math problems and sometimes achieve human-level performance (Cobbe et al., 2021b; Gao et al., 2024; Lin et al., 2024; Lightman et al., 2023; Zheng et al., 2024; Zhou et al., 2024). Despite these advances, the prevailing evaluation paradigm remains narrowly centered on **final-answer accuracy**, overlooking the step-by-step reasoning skills needed to reach a solution. This creates a critical **blind spot**: modelsthat guess, rely on spurious shortcuts, or produce superficially plausible steps are judged equally successful as models that follow valid logical progressions. This problem is especially acute in reasoning-intensive domains like mathematics, where problem solving unfolds through intermediate reasoning steps aligned with specific reasoning skills (e.g., applying formulas, setting up equations, performing logical deductions). These fine-grained reasoning skills form the fundamental units of LLMs' mathematical competence. Assessing performance at this granularity not only reveals whether a model solves a problem, but also how it arrives at the solution and where errors occur. This perspective enables the diagnosis of issues such as hallucinated steps, partial mastery of concepts, and logical inconsistencies that remain hidden under answer-only evaluation (Rahman et al., 2025; Ouyang, 2025; Bang et al., 2025).

To address this fundamental challenge, we propose **Res-Bench**, a first **Re**asoning **S**kill-aware evaluation benchmark derived from Chinese middle- and high-school level math problems. Each problem is annotated automatically by GPT-4 and the quality verification is conducted by human experts with its

---

[1]Code and data will be available upon publication.

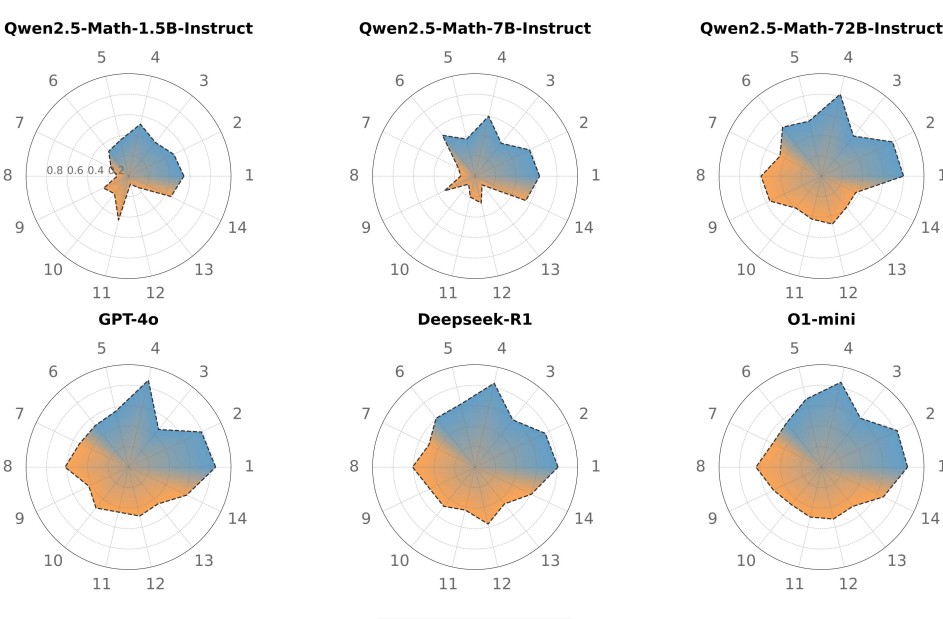

Figure 1: Overview of part of the evaluation results on the middle-school subset of RES-BENCH. The blue parts of each radar map are the performance of LLMs on understanding math problem while the orange parts of each radar map are the performance on solving math problems. See details of each reasoning skill in Table 6.

intermediate steps and the corresponding reasoning skills, enabling systematic step- and reasoning skill-level evaluation. Table 1 illustrates a data sample of the proposed Res-Bench, which contains four components: 1) **Question**, a middle or high-school level math problem; 2) **Steps**, intermediate solutions for reaching the correct final answer; 3) **reasoning skills**, the specific reasoning skill involves in each intermediate step, including two types: understanding the meaning of the problem and solving the problem; 4) **Answer**, the final answer to the given question. Based on Res-Bench, we further introduce a multi-dimensional evaluation framework that measures (i) final-answer accuracy, (ii) correctness of intermediate reasoning steps, and (iii) alignment with the required reasoning skills.

Our extensive experiments across a range of state-of-the-art LLMs, including 8 open-source LLMs (Qwen2.5 series models (Yang et al., 2024), Deepseek-R1 DeepSeek-AI et al. (2025)) and 2 commercial LLMs (GPT-4o and O1-mini (Achiam et al., 2023)), revealed a striking and consistent finding: a significant performance degradation when transitioning from final-answer accuracy to our proposed fine-grained metrics. This observation, which we term the "Accuracy-Fidelity Gap", shows that a model's ability to produce the correct final answer does not guarantee it has followed a valid or sound procedure. For example, the QwQ-32B-Preview model achieved an F1 score of 85.12% on final answers, but its performance dropped to 72.47% for intermediate steps and further to 66.87% for reasoning skill alignment.

As shown in Figure1, further analysis using our framework revealed that models consistently performed better on "understanding-based" skills, such as recognizing problem categories, than on "solving-based" skills like setting up equations or using logical inference. This suggests that LLMs are more proficient at semantic interpretation and pattern recognition than at the higher-level, strategic formulation of a solution. Furthermore, our analysis of step-by-step correctness demonstrated a performance gap based on model size; larger models were better able to maintain logical consistency over longer, more complex reasoning chains than their smaller counterparts.

In this paper, we propose a reasoning skill-level annotated dataset for mathematical reasoning, where each problem is decomposed into intermediate reasoning steps with explicit mappings to reasoning skills. This dataset serves as the foundation for our fine-grained evaluation framework that assesses LLMs not only on their ability to arrive at the correct final answer, but also on the soundness and completeness of their reasoning process as well as their alignment with relevant reasoning skills.

> **Question:** In the city math competition, schools A, B, and C respectively, have 1, 2, and 3 students winning the first prize. Arranging these six students in a line for a group photo requires that students from the same school are adjacent. The different arrangements total _______ ways.
>
> **Step 1:**
> Recognize that this is a permutation problem involving the arrangement of students in a line with specific constraints. Recognize the number of students from each school: School A has 1 student, School B has 2 students, and School C has 3 students, totaling 6 students.
>
> > ***Reasoning Skill:*** *require model to understand question: Recognize the problem category, such as algebra or geometry.*
>
> **Step 2:**
> Understand that students from the same school must be adjacent, which means each school's students can be treated as a single unit or block. Apply the bundling method to treat each school's students as a single unit. There are 3 units (schools) to arrange, and within each unit, the students can be arranged among themselves.
>
> > ***Reasoning Skill:*** *require model to solve the question: Apply appropriate mathematical formula or theorem.*
>
> **Step 3:**
> Set up the equation for the total number of arrangements: $A_3^3$ (arrangements of 3 units) * $A_3^3$ (arrangements within School C) * $A_2^2$ (arrangements within School B) * $A_1^1$ (arrangements within School A).
>
> > ***Reasoning Skill:*** *require model to solve the question: Set up equations or inequalities based on conditions.*
>
> **Step 4:**
> Calculate the permutations: $A_3^3 = 3! = 6$, $A_3^3 = 3! = 6$, $A_2^2 = 2! = 2$, $A_1^1 = 1! = 1$. Multiply these together: $6 * 6 * 2 * 1 =(\boxed{72})$.
>
> > ***Reasoning Skill:*** *require model to solve the question: Perform step-by-step calculations or manipulations.*
>
> **Answer:** 72

Table 1: An Example of Res-Bench with annotated intermediate steps and reasoning skills.

Extensive experiment results underscore the importance of process-based evaluation in advancing the development of LLMs toward interpretable and trustworthy reasoning systems.

## 2 RELATED WORK

As mentioned in the Introduction, dominant evaluation datasets focus on evaluating the ability of LLMs' math reasoning ability to correctly infer the correct answer and related steps. And it ignores the evaluation of the knowledge system required for solving problems by the large model.

### 2.1 MATH REASONING

In the realm of math reasoning, mathematical problem solving is a common benchmark for evaluating LLM reasoning capabilities. Widely used datasets such as GSM8K (grade-school math word problems) (Cobbe et al., 2021b) and MATH500 (competition-level problems with detailed solutions) have driven advances in chain-of-thought prompting and verification techniques (Hendrycks et al., 2021; Cobbe et al., 2021a). However, these benchmarks typically rely on the accuracy of the final answer as the primary evaluation signal. Even though some methods generate intermediate steps, the evaluation pipeline rarely assesses fine-grained reasoning fidelity or identifies which reasoning skills are actually being applied. There are also several existing benchmarks related to assessing

Table 2: Comparison between RES-BENCH and previous math reasoning benchmarks. [†]: Difficulty diversity denotes the diversity of problem types; our data varies from middle-school to high-school level math problems. [‡]: Data diversity denotes the diversity of problem types of the benchmarks. Our **Res-Bench** evaluates not only the performance of intermediate steps in the model, but also the model's mastery of relevant reasoning skills for each step, while balancing the diversity of data and difficulty.

| | Problem Diffculty[†] | Data Diversity[‡] | Step Annotation | Annotator | Reasoning Skill |
|---|---|---|---|---|---|
| CriticBench (Lin et al., 2024) | ★★ | ★★★ | ✗ | Human+Synthetic | ✗ |
| MathCheck-GSM (Zhou et al., 2024) | ★ | ★ | ✗ | Synthetic | ✗ |
| PRM800K (Lightman et al., 2023) | ★★ | ★ | ✓ | Human | ✗ |
| OMNI-MATH (Gao et al., 2024) | ★★★ | ★★★★ | ✗ | Human | ✗ |
| GSM8K (Cobbe et al., 2021b) | ★ | ★ | ✗ | Human | ✗ |
| AIME24 | ★★★★★ | ★ | ✗ | Human | ✗ |
| AMC23 | ★★★★ | ★ | ✗ | Human | ✗ |
| MATH500 | ★★ | ★★★★ | ✗ | Human | ✗ |
| ProcessBench (Zheng et al., 2024) | ★★★ | ★★★ | ✓ | Human | ✗ |
| RES-BENCH | ★★★ | ★★★★ | ✓ | Human+Synthetic | ✓ |

language models' reasoning process. Omni-Math and CriticBench (Gao et al., 2024; Lin et al., 2024) evaluate language models' abilities to critique solutions and correct mistakes in various reasoning tasks. MathCheck (Zhou et al., 2024) synthesizes solutions containing erroneous steps using the GSM8K dataset (Cobbe et al., 2021b), in which language models are tasked with judging the correctness of final answers or reasoning steps. PRM800K (Lightman et al., 2023) builds on the MATH problems (Hendrycks et al., 2021) and annotates the correctness and soundness of reasoning steps in model-generated solutions. It has also sparked a blooming of research interest in building process reward models (PRMs) (Wang et al., 2023). However, all of the previous datasets ignore the evaluation of LLMs' reasoning skills by assessing the intermediate steps with corresponding reasoning skills.

## 2.2 STEP WISE EVALUATION

Previous reasoning ability evaluation on LLMs often focuses exclusively on final-answer correctness. However, correct outputs can result from spurious shortcuts rather than sound reasoning. Notably, Wu et al. (2024) introduces CofCA, undertaking a detailed investigation of the LLMs' capabilities to reason on counterfactual passages. Their findings revealed that notable LLMs such as GPT-4(Achiam et al., 2023), Qwen(Bai et al., 2023), and LlaMA(Touvron et al., 2023) get inflated high performance and benefit from a high proportion of incorrect reasoning chains. To address this, researchers have proposed process-based evaluation approaches that assess intermediate reasoning steps(Yang et al., 2025; Thawakar et al., 2025). Chain-of-Thought (CoT) prompting encourages models to output multi-step reasoning traces, which has improved performance on complex tasks Wei et al. (2022). Enhancements such as self-consistency, which aggregates multiple reasoning paths, and verifier models, which score candidate solutions or sub-steps, further strengthen the reliability and robustness of reasoning (Wang et al., 2022; Cobbe et al., 2021a). While these methods emphasize correctness of intermediate steps, they often lack explicit mapping to underlying knowledge components—something our work addresses directly.

In contrast to the previous works, our proposed Res-Bench aims to objectively and realistically reflect the performance of LLMs in mathematical reasoning tasks by annotating fine-grained Intermediate steps and corresponding reasoning skills. The evaluation results of the performance of each intermediate step and corresponding reasoning skill reflect the real math reasoning ability of LLMs.

## 3 REASONING SKILL-AWARE EVALUATION

This section details the dataset construction process, including task definition, data annotation, quality verification, data statistics and evaluation protocols of the Res-Bench.

## 3.1 TASK DEFINITION

As shown in Table 1, given a math problem, step-by-step solutions, and corresponding reasoning skills, Res-Bench evaluates LLMs' performance on identifying the correct final answer, reasoning skill by assessing the intermediate steps. Formally, given a math problem $P$ with its step-by-step solution $S = \{s_1, s_2, ..., s_n\}$ and reasoning skills $K = \{k_1, k_2, ..., k_n\}$, the task is to output an correct answer $A$ with $s\_i$ and $k\_j$, $i \in \{-1, 0, ..., n\}$. $i = -1$ indicates that all steps are correct, $j = -1$ indicates that all steps are aligned with reasoning skills. Typically but non-inclusively, we consider a step as erroneous if it contains any of the following: (1) **Final Answer Errors**: incorrect final answers; (2) **Intermediate Step Errors**: invalid deductions, unwarranted or flawed reasoning steps. (3) **reasoning skill alignment errors**: misunderstanding or misapplication of mathematical or problem concepts.

## 3.2 DATA COLLECTION AND ANNOTATION

We collect over 5000 math problems from the Chinese middle school and high school level math problems in mathematical reasoning tasks. Inspired by recent studies on LLMs' ability to aid human annotation and avoid data contamination (Bartolo et al., 2021; Törnberg, 2023; Wu et al., 2024), we design a pipeline for automatically annotating math problems. Given a raw Chinese math problem, LLMs are required to act as translators to translate Chinese into English. Since LLMs are pre-trained on large scale corpus, to avoid data contamination, we prompt LLMs to randomly replace all the noun phrases, named entities of the translated math problem, and paraphrase it into a new math problem with numbers, variables, and concepts remaining unchanged.

Then we annotate solutions using the widely used commercial LLMs, GPT-4(Achiam et al., 2023) and the prompt is shown in Table 1. Given a math problem, LLMs are required to act as a math annotation assistant to generate the final as well as the intermediate steps, and corresponding reasoning skills. We manually defined 14 reasoning skill types that are used in middle- and high-school level problems, shown in Table 6.

**Expert Verification** We use LLMs to generate answers, steps, and reasoning skills to fit the given math problems. To make sure the final answers are correct and the steps and reasoning skills are related to each other, we use DeepSeek-R1 (DeepSeek-AI et al., 2025) and Qwen2.5-Math 72B-Instruct to verify the annotations, the correctness of final answers in the model-generated solutions against the reference answers. We follow the settings of ProcessBench (Zheng et al., 2024) and recruit five human experts with Chinese high school-level mathematical expertise for annotation, and all of them are required to conduct the majority vote for each annotation, and filter out the wrong annotations for the five experts can not reach consensus.

Table 3: Statistics of RES-BENCH. "% annotation errors" denotes the proportion of samples with *error annotation Intermediate steps* among all Intermediate steps with *correct final answers*. "% 3/n agreement" denotes the proportion of samples where the three-annotator agreement is achieved within $n$ annotators, so $(\% \ 3/3) + (\% \ 3/4) + (\% \ 3/5) = 100\%$. "% $\leq n$ steps" denotes the proportion of samples whose solutions have $\leq n$ steps .

| | Middle | | High | | Middle | | High | |
|---|---|---|---|---|---|---|---|---|
| | error-steps | correct-steps | error-steps | correct-steps | error-skills | correct-skills | error-skills | correct-skills |
| # Samples | 791 | 6235 | 967 | 6752 | 887 | 6139 | 1278 | 6441 |
| % annotation errors total annotations | $\frac{791}{7026} = 11.3\%$ | | $\frac{967}{7026} = 13.8\%$ | | $\frac{887}{7719} = 11.5\%$ | | $\frac{1278}{7719} = 16.6\%$ | |
| % 3/3 agreement | 72.3% | | 82.1% | | 62.3% | | 71.2% | |
| % 3/4 agreement | 20.1% | | 13.6% | | 21.4% | | 17.5% | |
| % 3/5 agreement | 7.6% | | 4.3% | | 16.3% | | 11.3% | |
| Distribution of Reasoning Steps | **Middle** | | | | **High** | | | |
| % $\leq$ 5 steps | 52.3% | | | | 48.7% | | | |
| % $\leq$ 10 steps | 31.7% | | | | 32.5% | | | |
| % $\leq$ 15 steps | 9.6% | | | | 11.4% | | | |
| % $\leq$ 20 steps | 6.4% | | | | 7.4% | | | |

## 3.3 EVALUATION

We evaluate models along three complementary axes—(1) final-answer correctness, (2) step-level reasoning correctness, and (3) reasoning-skill alignment—to reveal not only whether a model reaches the right answer but how it reaches it and which mathematical concepts it uses.

**Final-answer accuracy** For each test sample $i$, final-answer accuracy is computed as the exact match score between model-predicted answer $A_i^{pred}$ and reference answer $A_i^{ref}$.

This metric captures the standard end-to-end success rate and is reported separately for the middle- and high-school subsets. We use the exact match score for each sample.

**Step-level correctness** Each problem in Res-Bench is annotated with a sequence of reference intermediate steps $S = \{s_1, s_2, ..., s_n\}$. To evaluate model-generated solutions we align model-generated steps $\hat{S} = \{\hat{s}_1, \hat{s}_2, ..., \hat{s}_n\}$ with the reference steps in monotonic order (the $i-$th produced step is matched to the $t-$th reference step ). Unmatched steps (insertions/deletions when $m \neq n$ are treated as incorrect for the purposes of step-level scoring. From the aligned steps, we compute:

- **Step-level accuracy (micro):**

$$\text{Acc}_{\text{step}} = \frac{\sum_{i=1}^{N} \sum_{t=1}^{\tilde{n}_i} \mathbf{1}\{\hat{s}_{i,t} \text{ is correct}\}}{\sum_{i=1}^{N} \tilde{n}_i}, \tag{1}$$

  where $\tilde{n}_i$ is the number of aligned steps for sample $i$.

- **All-steps-correct rate (sample-level):** the fraction of samples for which every aligned step is judged correct.

- **Earliest-error localization accuracy:** for each sample we identify the earliest erroneous step index in the annotations $e_i^{\text{ref}}$ and the earliest erroneous step in the model output $e_i^{\text{pred}}$. We report the proportion of samples with exact matches $e_i^{\text{pred}} = e_i^{\text{ref}}$ and provide a $\pm 1$ tolerance variant to account for minor segmentation differences.

**Reasoning Skill Alignment** Res-Bench associates each reference step with an explicit reasoning skill index drawn from the taxonomy listed in Table 6. For every aligned model step $\hat{s}_{i,t}$ we require the model to indicate (or be mapped to) a single reasoning skill index $\hat{k}_{i,t} \in \mathcal{K}$ that best describes the mathematical concept or operation used in that step (e.g., "set up equations", "apply formula", "simplify expressions"). We evaluate the correctness of the model's selection in two ways:

- **Reasoning skill selection accuracy (top-1):**

$$\text{Acc}_{\text{kp}} = \frac{\sum_{i=1}^{N} \sum_{t=1}^{\tilde{n}_i} \mathbf{1}\{\hat{k}_{i,t} = k_{i,t}^{\text{ref}}\}}{\sum_{i=1}^{N} \tilde{n}_i}, \tag{2}$$

  where $k_{i,t}^{\text{ref}}$ is the annotated reasoning skill for sample $i$'s $t$-th step.

- **Per-KP precision/recall/F1:** to reveal which reasoning skills models master or confuse, we compute precision, recall, and F1 for each reasoning skill and report macro- and micro-averaged values.

When a model does not explicitly output a reasoning skill index, we obtain its implicit reasoning skill labels by prompting it to annotate each previously generated step with the kp list; if automatic mapping remains ambiguous we fall back to human adjudication and exclude highly ambiguous cases from Res-accuracy aggregation.

**Aggregation and summary statistics** All metrics are computed separately for the middle- and high-school subsets and then aggregated as needed. Because model behavior on error vs. correct samples can differ substantially, we also compute accuracy on (a) samples whose *final answer* is correct and (b) samples whose final answer is incorrect; to summarize model robustness we report the harmonic mean (F1) of these two accuracies where appropriate. In addition, we analyze distributions of kp performance (which reasoning skills are most/least well selected) and error patterns across step

positions (early vs. late steps). The fine-grained nature of these metrics makes Res-Bench suitable both for evaluating end-to-end solving ability and for diagnosing *how* and *why* a model's reasoning succeeds or fails. **Note:**The reasoning skill taxonomy used for alignment and scoring is the 14-point list shown in Table 6.

## 4 EXPERIMENTS

We conduct extensive experiments to evaluate the mathematical reasoning capabilities of advanced LLMs across three dimensions: final answers, intermediate steps, and reasoning skill alignment. Specifically, our study seeks to address the following research questions:(1) **Reasoning Ability**: To what extent can LLMs produce correct final answers while also generating valid intermediate reasoning steps? (2) **reasoning skill**: Which specific reasoning skills are used and exhibited by LLMs during the process of solving mathematical problems? (3) **Alignment**: How well can LLMs align their step-by-step reasoning with the appropriate reasoning skills required for problem solving?

### 4.1 EXPERIMENT SETTINGS

**Metrics**   We evaluate models on Res-Bench along the three complementary axes introduced in §3: (1) final-answer accuracy, (2) step-level correctness, and (3) reasoning skill alignment. Concretely, for each Res-Bench subset (middle-school and high-school) we compute:

- **Final-answer accuracy** $\text{Acc}_{\text{answer}}$ as the proportion of samples whose predicted final answer equals the reference. :contentReference[oaicite:0]index=0
- **Step-level metrics:** (i) step-level accuracy (micro) $\text{Acc}_{\text{step}}$ computed over all aligned steps, (ii) all-steps-correct rate (sample-level), and (iii) earliest-error localization accuracy (with a $\pm 1$ tolerance variant).
- **Reasoning skill metrics:** top-1 reasoning skill selection accuracy $\text{Acc}_{\text{kp}}$ (per-step), and per-KP precision / recall / F1 (reported as macro- and micro-averages). When a model does not explicitly emit KP labels we prompt it to annotate its generated steps; ambiguous mappings are adjudicated by humans and excluded from aggregated kp metrics (these exclusions are tracked).
- **Balanced comparison (F1):** for the primary model comparison we follow Res-Bench and primarily use the harmonic mean (F1) of accuracy on erroneous vs. correct-sample subsets to balance models that are overly conservative vs. overly permissive.

All reported metrics are computed separately for the middle- and high-school subsets and then aggregated as needed (we additionally analyze per-step-position error distributions and per-KP performance breakdowns).

**Models**   We present results from a range of state-of-the-art (SoTA) proprietary LLMs, including OpenAI's GPT-4 (Achiam et al., 2023), Deepseek-R1 (DeepSeek-AI et al., 2025), and O1-mini (El-Kishky, 2024). Regarding open-source models, we consider Instruct version models of LLaMA-3 (Touvron et al., 2023), Qwen2.5 (Yang et al., 2024).

### 4.2 RESULTS

We present the evaluation results of multiple state-of-the-art LLMs on Res-Bench across three dimensions: final-answer accuracy, intermediate-step correctness, and reasoning skill alignment. The results are summarized in Table 4, with F1 scores reported for both middle-school and high-school subsets, as well as averaged across all metrics.

**The "Accuracy-Fidelity" Gap**   The data in Table 4 reveals a striking and consistent pattern: for every model evaluated, a significant performance drop is observed when moving from the final-answer F1 score to the intermediate-step and reasoning-skill F1 scores. For instance, QwQ-32B-Preview achieves a high final-answer score of 85.12% on the middle-school subset. However, its performance on intermediate steps drops to 72.47%, and its reasoning skill alignment score falls even further to 66.87%. This pattern is not an isolated observation; it is a fundamental characteristic of LLM

performance across the board. This consistent decay is a profound finding because it demonstrates that a model's ability to produce a correct final answer does not guarantee that it followed a sound or valid procedure to get there. The data suggests that a model's correct output may be the result of a "spurious shortcut" or even a "hallucinated" set of steps that happens to lead to the right conclusion.

**Intermediate-Step Correctness**   Step-level reasoning remains a challenge for all models. While final-answer accuracy is relatively high, step-level F1 scores are consistently lower. For example, QwQ-32B-Preview achieves 72.47% on intermediate steps, still leading among open-source models but trailing behind o1-mini (62.34%) and GPT-4o (60.75%). This suggests that even when models produce correct final answers, their reasoning processes often contain errors or inconsistencies. The earliest-error localization accuracy (not shown in Table 4 but implied by the evaluation protocol) further reveals that models frequently make mistakes in early reasoning steps, which propagate and invalidate subsequent reasoning.

Figure 2 illustrates the distribution of correct intermediate steps for two Qwen2.5 models (Math-1.5B and Math-7B). The results provide a crucial insight: as the number of reasoning steps increases, a significant performance gap emerges between models of different scales. The smaller 1.5B model shows a sharp decline in the proportion of fully correct step sequences as the number of steps grows. In contrast, the 7B model maintains a relatively higher level of correctness over longer reasoning chains. This indicates that larger models possess a superior ability to maintain coherence and logical consistency throughout extended, multi-step reasoning processes. The performance gap is most pronounced for problems requiring more than 10 steps, highlighting that smaller models struggle with the sustained focus and complex dependency management required for lengthy derivations, while larger models demonstrate more robust reasoning capabilities.

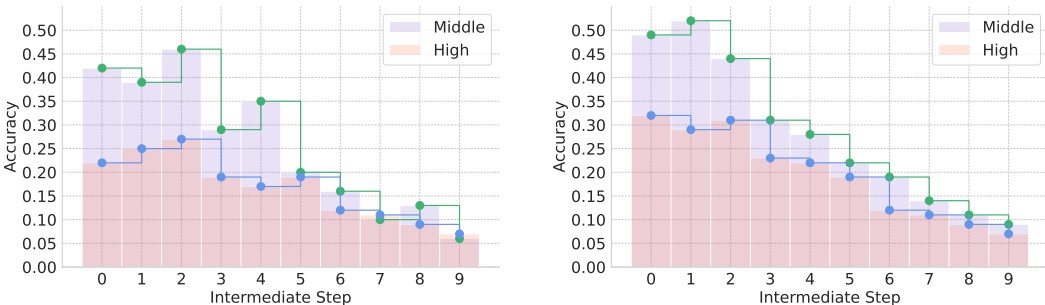

Figure 2: Left: Distribution of correct intermediate steps of Qwen2.5-1.5B-Instruct. Right: Distribution of correct intermediate steps of Qwen2.5-7B-Instruct.

**Per-reasoning skill Performance**   Based on the reasoning skill taxonomy in Table 6 and the overview in Figures 1 and 3, we observe that models perform better on understanding-bases reasoning skills (e.g., Skill 1: Extract numerical values and units from the problem statement, Skill 4: Recognize the problem category such as algebra or geometry) than on solution-based points (e.g., Skill 7: Apply appropriate mathematical formula or theorem, Skill 9: Set up equations or inequalities based on conditions). This suggests that LLMs are more proficient in executing known procedures than in formulating problem-solving strategies.

This pattern provides a critical insight into the nature of LLM reasoning. A model's strength in understanding skills suggests that it excels at semantic interpretation and pattern recognition from the problem statement. It can effectively read a problem, identify the key components, and recall a relevant schema from its training data. However, its weakness in solving skills—particularly those requiring strategic formulation like "Set up equations" or "Use logical inference"—indicates a deficiency in its ability to perform novel, strategic planning. The models are adept at executing known procedures, such as performing calculations or applying a specific formula, but they struggle with the higher-level, generative process of constructing a multi-step solution from scratch. The low reasoning skill alignment scores across the board further confirm that models often misapply or misidentify the required mathematical concepts, even if a step appears to be correct.

Table 4: Evaluation results on RES-BENCH. We report the F1 score of the respective accuracies on erroneous and correct samples. Models are evaluated with zero-shot chain-of-thought prompting.

| Model | Middle | High | Intermediate-Steps | Reasoning-Skill | Average |
|---|---|---|---|---|---|
| *Llama Series* | | | | | |
| Meta-Llama-3-8B-Instruct | 44.67 | 13.12 | 15.48 | 10.15 | 20.86 |
| Llama-3.1-8B-Instruct | 42.34 | 11.75 | 10.45 | 9.67 | 18.56 |
| Llama-3.1-70B-Instruct | 59.88 | 29.54 | 35.67 | 30.14 | 38.81 |
| Llama-3.2-1B-Instruct | 35.85 | 6.72 | 7.88 | 6.12 | 14.14 |
| Llama-3.2-3B-Instruct | 38.77 | 9.34 | 12.46 | 7.89 | 17.11 |
| Llama-3.3-70B-Instruct | 60.12 | 30.32 | 37.88 | 29.41 | 39.43 |
| *Qwen Series* | | | | | |
| Qwen2.5-Math-1.5B | 42.14 | 13.59 | 14.12 | 9.27 | 19.78 |
| Qwen2.5-Math-1.5B-Instruct | 44.89 | 16.22 | 18.34 | 11.22 | 23.42 |
| Qwen2.5-Math-7B | 46.71 | 18.34 | 20.48 | 17.56 | 25.77 |
| Qwen2.5-Math-7B-Instruct | 51.91 | 21.76 | 25.92 | 21.87 | 30.37 |
| Qwen2.5-Math-72B-Instruct | 68.91 | 39.78 | 45.88 | 36.48 | 47.76 |
| Qwen2.5-1.5B-Instruct | 40.01 | 12.69 | 15.42 | 9.65 | 19.44 |
| Qwen2.5-7B-Instruct | 47.73 | 17.89 | 19.37 | 15.44 | 25.11 |
| Qwen2.5-14B-Instruct | 51.33 | 20.12 | 23.52 | 20.14 | 28.78 |
| Qwen2.5-32B-Instruct | 54.55 | 24.36 | 32.55 | 27.89 | 34.84 |
| Qwen2.5-72B-Instruct | 64.56 | 34.92 | 41.56 | 32.33 | 43.34 |
| ★ **QwQ-32B-Preview** | **85.12** | **69.33** | **72.47** | **66.87** | **73.45** |
| *Proprietary LLMs* | | | | | |
| GPT-4o | 74.67 | 45.23 | 60.75 | 52.38 | 58.26 |
| Deepseek-R1 | 71.45 | 43.47 | 59.87 | 50.14 | 56.23 |
| o1-mini | 77.57 | 48.95 | 62.34 | 55.47 | 61.08 |

## 5 CONCLUSION

We introduce Res-Bench, a novel benchmark designed for the fine-grained evaluation of mathematical reasoning in Large Language Models (LLMs). Moving beyond the standard metric of final-answer accuracy, Res-Bench provides a multi-dimensional assessment framework that measures a model's proficiency in generating valid intermediate reasoning steps and its ability to align these steps with the correct underlying reasoning skills. Our extensive evaluation of state-of-the-art models reveals a critical discrepancy: while LLMs can often produce correct final answers, their step-by-step reasoning processes frequently contain logical inconsistencies, errors, and a misapplication of the required mathematical concepts. This underscores the insufficiency of answer-only evaluation and highlights the necessity of process-based assessment to truly understand and advance the reasoning capabilities of LLMs. We envision Res-Bench serving as a foundational tool for the community, enabling more robust diagnosis of model weaknesses, guiding the development of models that reason more faithfully, and ultimately driving progress toward more interpretable and trustworthy AI systems for mathematical problem-solving and beyond.

## ETHICS STATEMENT

All authors affirm their adherence to the ICLR Code of Ethics. We have carefully considered the ethical implications of our research, particularly concerning the safe and responsible deployment of Large Language Model (LLM)s. Our work directly addresses the critical need to avoid harm by mitigating risks such as dangerous diagnostic medical recommendations, financial losses, and privacy breaches, which can arise from the unconstrained operation of LLMs. We believe our work

contributes positively to human well-being by enhancing the safety and trustworthiness of advanced AI systems.

## REPRODUCIBILITY STATEMENT

The dataset used in this work (Res-Bench) is introduced and described in Section 3.2 of the paper. For all experiments involving Res-Bench, a complete description of the data processing steps—including problem translation, paraphrasing, step/reasoning skill annotation via GPT-4, and expert verification protocols—is provided in section 3.2. Part of the data will be uploaded as supplementary materials.

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

## LIMITATIONS

While Res-Bench provides a valuable framework for fine-grained evaluation of mathematical reasoning, our work has several limitations that point to directions for future research.

**Scope and Generalizability of the Dataset.** The problems in Res-Bench are primarily derived from Chinese middle- and high-school mathematics curricula. While this provides a focused domain for analysis, it may limit the generalizability of our findings to mathematical reasoning problems from other educational systems, cultures, or more advanced domains (e.g., undergraduate-level mathematics). The benchmark's effectiveness for evaluating reasoning in highly abstract or proof-based problems remains to be seen.

**Fixed Taxonomy of reasoning skills.** Our evaluation relies on a pre-defined taxonomy of 14 reasoning skills. Although this taxonomy was carefully designed to cover common operations in secondary school math, it is inherently non-exhaustive. This fixed set may not perfectly capture all nuances of reasoning for every problem, potentially leading to forced or ambiguous alignments for steps that involve composite or novel reasoning patterns not explicitly listed.

**Challenges in Step Alignment and Evaluation.** Evaluating step-level correctness and reasoning skill alignment requires aligning model-generated reasoning traces with the reference solution. This process can be challenging due to the variability in how different models express the same logical step. Although we employ human adjudication for ambiguous cases, the alignment process may not be perfectly robust to stylistic differences, potentially affecting the precision of step-level and reasoning skill metrics.

## A  THE USE OF LARGE LANGUAGE MODELS

We employed LLMs for grammar checking and polishing the English expression throughout this manuscript. It is important to note that while our research focuses on leveraging LLMs for data annotation and evaluation, the LLMs studied in this work are the subject of our research rather than tools for research ideation or scientific writing. All experimental design, analysis, and scientific conclusions were developed independently by the authors.

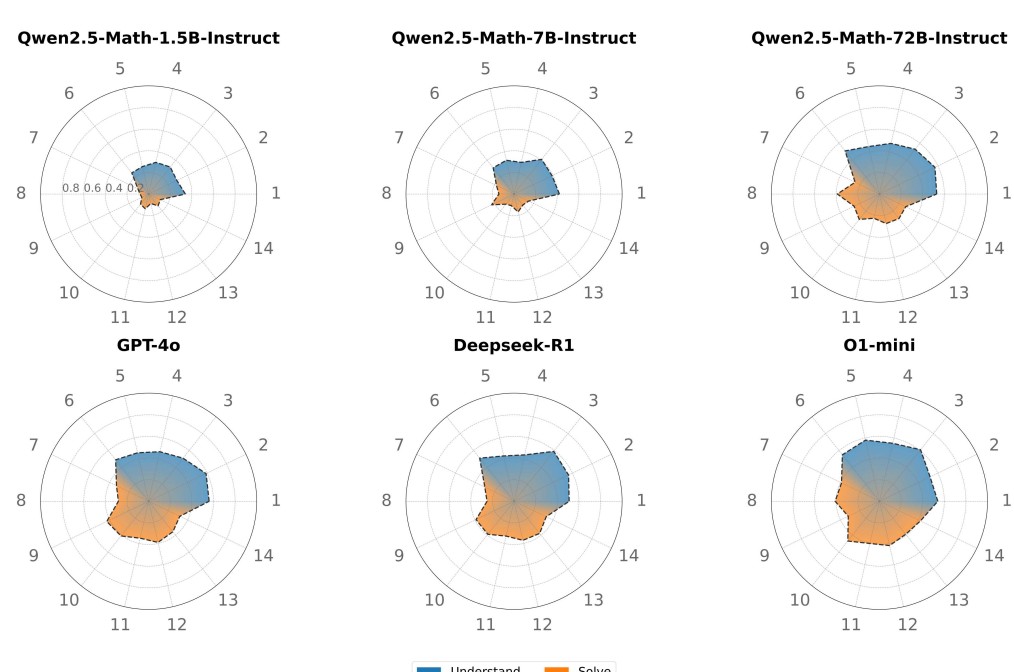

Figure 3: Overview performance of reasoning skills on the high-school subset of **Res-Bench**.

## B  POST TRAINING

To investigate the impact of specialized training on mathematical reasoning and reasoning skill alignment, we conducted a series of post-training experiments on the Qwen2.5-Math-1.5B and Qwen2.5-Math-7B models.

**Data Synthesis**    We synthesized an additional 1,000 high-quality math problems to augment the training data. This was done by prompting GPT-4 to generate middle-school and high-school level problems that mirror the style and complexity of Res-Bench. Each generated sample includes the question, a detailed step-by-step solution, and the corresponding reasoning skill for each step, following the annotation standard of our benchmark.

**Experimental Setup**    All post-training experiments were conducted on a cluster of 8 NVIDIA A100 GPUs. We maintained consistent hyperparameters across models to ensure a fair comparison, using a learning rate of 1e-5, a batch size of 32, and training for 3 epochs. The models were evaluated on the Res-Bench test set immediately after each training phase.

**Analysis of Post-Training Results**    The results of the post-training experiments are summarized in Table 5. The analysis reveals several key findings:

Significant Performance Gains: All post-training algorithms lead to substantial improvements over the base models (whose performance is shown in Table 4). For instance, the Qwen2.5-Math-7B model's average F1 score improved from 25.77% (base) to 36.32% after SFT, and further to 45.42% after GRPO. This demonstrates that targeted training on reasoning skill-annotated data is highly effective.

Progressive Improvement with Advanced Algorithms: The results show a clear hierarchy: GRPO > DPO > SFT. This indicates that while supervised fine-tuning (SFT) provides a strong baseline, algorithms that incorporate preference learning (DPO) and more sophisticated policy optimization (GRPO) are more effective at teaching the model not just what the correct steps are, but also how to select valid reasoning paths over invalid ones, leading to more robust reasoning.

| Model | Algorithm | Middle | High | Intermediate-Steps | Reasoning-Skill | Average |
|---|---|---|---|---|---|---|
| Qwen2.5-Math-1.5B | SFT | 53.71 | 20.34 | 24.56 | 15.47 | 28.52 |
| | DPO | 58.92 | 23.57 | 26.72 | 17.81 | 31.76 |
| | GRPO | 62.14 | 28.31 | 32.35 | 21.44 | 36.11 |
| Qwen2.5-Math-7B | SFT | 64.25 | 26.41 | 29.32 | 25.31 | 36.32 |
| | DPO | 66.71 | 28.41 | 32.14 | 27.33 | 38.65 |
| | GRPO | 71.26 | 36.44 | 38.57 | 35.41 | 45.42 |

Table 5: Performance of post-training on different algorithms.

Scalability with Model Size: The absolute performance and the relative gains from post-training are more pronounced for the 7B model compared to the 1.5B model. For example, GRPO led to an 18.1-point absolute improvement in the average score for the 7B model, compared to a 16.59-point improvement for the 1.5B model. This suggests that larger models have a greater capacity to absorb and benefit from the nuanced, structured knowledge provided by our training data.

Holistic Improvement Across Metrics: The improvements are consistent across all evaluation dimensions—final answer, intermediate steps, and reasoning skill alignment. This confirms that our post-training approach enhances the model's reasoning capabilities in a holistic manner, rather than overfitting to a single metric. The notable improvement in reasoning skill scores (e.g., from 17.56% to 35.41% for the 7B model) is particularly significant, as it shows that the models are learning to better associate reasoning steps with the correct underlying mathematical concepts.

In conclusion, the post-training experiments validate that Res-Bench is not only a diagnostic tool but also a valuable resource for model improvement. The structured, reasoning skill-aware data enables significant gains in reasoning quality, with advanced algorithms like GRPO yielding the most substantial benefits, especially for larger models.

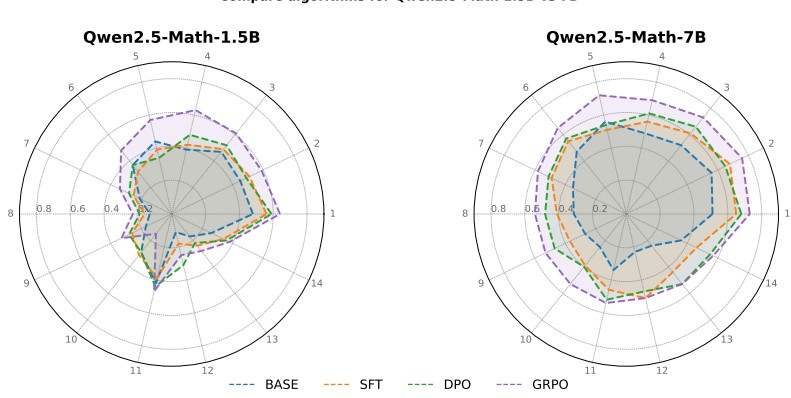

Figure 4: Overview of reasoning skills on the middle-school subset of **Res-Bench**.

**Instruction:**
You are an expert at high-school level math problem annotation. Given the Question, Solution, Please think step-by-step and generate fine-grained intermediate steps. For each intermediate step, assign exactly one corresponding reasoning skill chosen only from the following 14 reasoning skills (do not paraphrase or add new labels). Please output your reply in the following JSON format:
=====================================================================
Here are 14 reasoning skills:

- 1. require model to understand question: Extract numerical values and units from the problem statement.
- 2. require model to understand question: Extract given conditions and constraints.
- 3. require model to understand question: Identify the unknown quantity to be found.
- 4. require model to understand question: Recognize the problem category such as algebra or geometry.
- 5. require model to understand question: Understand relationships between given quantities.
- 6. require model to understand question: Interpret textual descriptions into mathematical expressions.
- 7. require model to solve the question: Apply appropriate mathematical formula or theorem.
- 8. require model to solve the question: Set up equations or inequalities based on conditions.
- 9. require model to solve the question: Perform step-by-step calculations or manipulations.
- 10. require model to solve the question: Simplify expressions through algebraic operations.
- 11. require model to solve the question: Solve equations or systems of equations.
- 12. require model to solve the question: Evaluate expressions to obtain numerical results.
- 13. require model to solve the question: Verify solution against given constraints.
- 14. require model to solve the question: Use logical inference for deductions or proofs.

=====================================================================
Here, I provide you with an example of the math data to help you understand your task.
First, I provide you with a Question:
[EXAMPLE QUESTION]
The solution of the Question is [EXAMPLE Solution], you should output:
[EXAMPLE RESPONSE]

Box 1: The prompts of annotating a math problem with reasoning skills and intermediate steps.

Table 6: List of annotated mathematical reasoning skills.

| ID | reasoning skill Description | Category |
|----|---------------------------|----------|
| 1 | Extract numerical values and units from the problem statement | Understand Question |
| 2 | Extract given conditions and constraints | Understand Question |
| 4 | Recognize the problem category such as algebra or geometry | Understand Question |
| 5 | Understand relationships between given quantities | Understand Question |
| 6 | Interpret textual descriptions into mathematical expressions | Understand Question |
| 7 | Apply appropriate mathematical formula or theorem | Solve Question |
| 8 | Perform step-by-step calculations or manipulations | Solve Question |
| 9 | Set up equations or inequalities based on conditions | Solve Question |
| 10 | Simplify expressions through algebraic operations | Solve Question |
| 11 | Solve equations or systems of equations | Solve Question |
| 12 | Evaluate expressions to obtain numerical results | Solve Question |
| 13 | Verify solution against given constraints | Solve Question |
| 14 | Use logical inference for deductions or proofs | Solve Question |

