# OpenReview forum: "Res-Bench: Reasoning Skill-Aware Reasoning Diagnostic evaluation benchmark for math reasoning"
_ICLR.cc/2026/Conference — ICLR 2026 Conference Withdrawn Submission_

### Official Review · Reviewer_5gxB · 2025-10-27

**Soundness:** 1
**Presentation:** 2
**Contribution:** 1
**Rating:** 2
**Confidence:** 5

**Summary:**

This paper introduces Res-Bench, a math reasoning dataset with 3271 test samples,
The authors claim that Res-Bench is characterized by process-level evaluation and skill-aware evaluation.
They conduct experiments on Res-Bench using Qwen-series models and llama-series models, as well as three commercial LLMs.
The main finding is that although LLMs have a high final answer accuracy, they are prone to having incorrect steps during reasoning.

**Strengths:**

The proposed Res-Bench is claimed to be verified by human experts, and in some sense might be useful for related domains.

**Weaknesses:**

1. The idea of process-level evaluation is not novel and has been well-studied in the past 2 years since OpenAI's work: Let's Verify Step by Step! And many important related works are not covered in this work. To name a few: ROSCOE [1]. VerifyBench [2], [3].

2. Step-level evaluation, in essence, is very hard to define, because how to segment a solution into different steps will significantly influence final results, especially for large reasoning models whose responses are extremely long and contain many reflection behaviors.

3. In L235-L238, the authors discuss the data contamination issue, but I think it is not compelling enough. Data contamination tools like [4] are suggested.

4. The human-designed skills are vague, and the detailed definitions are not given. Even human experts may struggle to classify a step into these skills provided in the Appendix. I do not believe evaluation based on skills is reliable.

5. Experiments are not comprehensive enough; more LLMs across different model families are needed, not solely on Qwen and Llama Series.

6. Important evaluation details are not given, e.g, the maximum output length, decoding parameters.

7. In Table 4, the results are in some sense weird: QwQ-32B-preview beats DeepSeek-R1 and O1-mini by a large margin. I suggest that the authors check the API returns to see whether there are bugs in their evaluation scripts. Besides, the extremely high QwQ-32B-preview results may suggest a data contamination issue.

8. Others: L240: wrongly referred to Table 1. L90: not striking at all, just common sense now.

[1] Golovneva, O., Chen, M., Poff, S., Corredor, M., Zettlemoyer, L., Fazel-Zarandi, M., & Celikyilmaz, A. (2022). Roscoe: A suite of metrics for scoring step-by-step reasoning. arXiv preprint arXiv:2212.07919.

[2] Yan, Y., Jiang, J., Ren, Z., Li, Y., Cai, X., Liu, Y., ... & Zhuang, Y. (2025). VerifyBench: Benchmarking Reference-based Reward Systems for Large Language Models. arXiv preprint arXiv:2505.15801.

[3] Xia, S., Li, X., Liu, Y., Wu, T., & Liu, P. (2025, April). Evaluating mathematical reasoning beyond accuracy. In Proceedings of the AAAI Conference on Artificial Intelligence (Vol. 39, No. 26, pp. 27723-27730).

[4] Xu, R., Wang, Z., Fan, R. Z., & Liu, P. (2024). Benchmarking benchmark leakage in large language models. arXiv preprint arXiv:2404.18824.

**Questions:**

See Weakness.

---

### Official Review · Reviewer_khby · 2025-10-28

**Soundness:** 2
**Presentation:** 3
**Contribution:** 2
**Rating:** 4
**Confidence:** 4

**Summary:**

This paper introduces Res-Bench, a new diagnostic benchmark for mathematical reasoning evaluation. The motivation is that current evaluations mainly focus on final answer accuracy, which fails to reflect whether models truly reason correctly. The authors propose a multi-dimensional evaluation framework that jointly examines: Final Answer accuracy, Intermediate Step correctness Reasoning Skill alignment. Res-Bench includes 3,271 math problems aligned with Chinese middle– and high-school curricula, annotated first by GPT-4 and then verified by human experts. Experiments across various LLMs indicate a clear gap between answer accuracy and reasoning soundness — termed the “Accuracy–Fidelity Gap”. The dataset reveals that models often hallucinate or misuse reasoning skills even when final answers are correct.

**Strengths:**

1.The benchmark evaluates models simultaneously on final outputs, step-level correctness, and reasoning skill mastery，and successfully identifies nontrivial performance gaps across these dimensions.

2. A human-involved verification process ensures higher factual accuracy and reliability of the annotated dataset.

**Weaknesses:**

1.Data statistics require more clarity and completeness:  It’s unclear what types of problems are included (e.g., which math domains, whether proof problems exist, distribution across skill categories). Since the dataset is a key contribution, the paper should describe these aspects more rigorously.

2.Model coverage is limited:  Only three proprietary models are evaluated, and the range of model types is narrow. Adding more diverse systems would strengthen claims about generality and comparative insights.

3. Analysis and follow-up experiments are relatively basic:
The dataset enables deeper investigations that are not fully explored. For example: (1) What factors cause hallucination in reasoning skill usage? (data? training stage?) (2) Are there effective methods to consistently improve performance across the three evaluation metrics? e.g., some reliable reasoning techniques or PRM models. The authors can test them.

**Questions:**

1. Could the authors provide clearer statistics and descriptions of: Problem categories and knowledge points and whether proving questions are included

2. What are the results of different model families? Could adding more closed-source  / math-specialized / process reward models reveal different strengths across the three metrics?

---

### Official Review · Reviewer_7w3e · 2025-10-30

**Soundness:** 2
**Presentation:** 3
**Contribution:** 2
**Rating:** 4
**Confidence:** 3

**Summary:**

This paper introduces Res-Bench, which focuses on the step-level evaluation of LLM reasoning. Res-Bench covers middle to high school mathematics, provides annotations for step-level reasoning as well as final solutions of 3271 questions. The authors then propose to evaluate step-level reasoning correctness and reasoning skill alignment, which aims to identify incorrect reasoning traces or misclassification of reasoning skills. The paper is well motivated, and the experiments are detailed.

**Strengths:**

1. The paper is well-motivated.
2. The data annotation process is provided in detail, which helps readers understand the construction of Res-Bench.
3. The experiments cover a diverse collection of mainstream LLMs, both proprietary and open-source, which help shape the landscape of step-level reasoning among current LLMs.

**Weaknesses:**

1. Although the paper is motivated by the idea of step-level evaluation, it does not provide enough justifications for this motivation. For example, how often do models reach a correct answer with invalid intermediate steps? On what kind of questions/benchmarks do this phenomenon tend to occur? Does it occur on popular math benchmarks? The motivation would be far more convincing if the authors provide evidence that existing (math) benchmarks fail in this aspect.
1. The writing of the paper can be improved. For example, the notation $\hat s_{i,t}$ in Eq.(1) is quite confusing: is $i$ the step index discussed in line 283, or is it the sample index discussed in line 291?
2. The definition of Step-level correctness is very restrictive, which is also discussed in the Limitations section. The authors judge a model-generated step as correct if it matches the corresponding step in the reference answer. One would argue that reasoning traces may well be non-unique. Furthermore, requiring the model to reason using specific patterns (e.g. by giving an explicit step index) might undermine the model's reasoning ability, which does not seem like a fair evaluation method.
3. Details on how to align model-generated steps to reference steps are missing. I believe this is an essential component which directly affects the validity of the proposed step-level evaluation.
4. The motivation and evaluation protocol of Reasoning-Skill Alignment is somewhat unclear. I assume that the goal of reasoning skill alignment is to:
    (i) detect whether the reasoning model can utilize targeted reasoning skills.
    (ii) detect whether the reasoning model can correctly classify the reasoning skills used in its own reasoning steps.
    For point (i), this seems to be overlapped with Step-level correctness. For point (ii), one would argue that a clear understanding of reasoning skill taxonamy is not necessary for the correctness of the reasoning process itself. Failing to classify the correct reasoning skill does not mean the model fails to utilize that skill in its reasoning, which challenges the necessity of point (ii). This evaluation method also challenges the reasoning model's instruction following ability, which could potentially introduce confounding factors, since the model may fail to perform classification correctly simply due to lack of instruction following ability over long reasoning contexts.
5. The Post-Training section in Appendix B is quite sketchy. The authors should be more clear on the training objective of DPO and GRPO, particularly the reward design of GRPO. Moreover, this section does not seem to add much value. It's major claim is that models perform better on Res-Bench when trained on data similar to Res-Bench, which is not surprising. In my opinion, the authors should instead show that models trained with step-level annotation can generalize to other benchmarks, such that step-level correctness increases. But this also deviates from the main idea of the paper, which is to address the lack of step-level evaluation.

**Questions:**

1. Is there any evidence or pilot study that shows current reasoning models could achieve correct final answer with invalid intermediate steps, particularly on math benchmarks? Is this behaviour so often that it is indeed a concern?
2. How often is the step-level annotation of a given question non-unique? And if questions tend to have non-unique reasoning steps, how does that affect the validity of model's performance on Res-Bench?
3. What is the purpose of Reasoning Skill Alignment? Why not use a separate model to classify the reasoning skills used in the model's reasoning trajectory?

---

### Official Review · Reviewer_U6J7 · 2025-11-01

**Soundness:** 3
**Presentation:** 2
**Contribution:** 2
**Rating:** 2
**Confidence:** 4

**Summary:**

This paper introduces Res-Bench, a novel benchmark for evaluating the mathematical reasoning of Large Language Models (LLMs) beyond final-answer accuracy. Comprising 3,271 middle- and high-school level math problems, Res-Bench provides fine-grained annotations for intermediate reasoning steps and maps each step to an explicit reasoning skill (e.g., understanding the problem, applying formulas). Using a multi-dimensional evaluation protocol that assesses final answers, step-level correctness, and reasoning skill alignment, the study reveals a significant "Accuracy-Fidelity Gap": while models like GPT-4o and QwQ-32B often produce correct final answers, their step-by-step reasoning is frequently inconsistent and misaligned with the required skills. The findings underscore the limitations of answer-only evaluation and highlight the necessity of process-based assessment to develop more interpretable and trustworthy reasoning systems.

**Strengths:**

1.The proposal for the detection of intermediate steps and reasoning abilities is novel and has significant reference value for subsequent work.

2.The charts are beautifully made, which can well display the data and effectively illustrate the content.

**Weaknesses:**

1. In the "Expert Verification" part of section 3.2, what are the details about Qwen2.5-Math 72B-Instruct and DeepSeek-R1 used for verification? Specifically, what prompts are used? How is the consistency of the annotated text verified among five annotators?

2. In the "Step-level correctness" section of the "3.3 EVALUATION" part, I have some questions about the evaluation metric.

2.1. How is the evaluation conducted with your data in the "All-steps-correct rate (sample-level)" part? Was a large language model used, and if so, which one, and what’s the prompt?

2.2. The paper defines the metric - "Earliest-error localization accuracy". However, this metric wasn’t used to evaluate the main results?

2.3. Regarding the Per-KP precision/recall/F1 section, why wasn't the same metric used for the Step-level correctness part?

3.  The "RESULTS" section requires additional experimental data. Both the "middle" and "high" sections should include intermediate-step and reasoning skills to support your viewpoint of the “Accuracy-Fidelity” Gap.

**Questions:**

In the annotations of Table 3, it is mentioned that (% 3/3) + (% 3/4) + (% 3/5) = 100%. What are the specific meanings of three-annotator agreement? Does it refer to inter-annotator agreement? Therefore, it should be “% n”, which has nothing to do with 3.

---

### Note · Authors · 2025-11-13

I have read and agree with the venue's withdrawal policy on behalf of myself and my co-authors.